# CAN ALTQ LEARN FASTER: EXPERIMENTS AND THEORY

## ABSTRACT

Differently from the popular Deep Q-Network (DQN) learning, Alternating Q-learning (AltQ) does not fully fit a target Q-function at each iteration, and is generally known to be unstable and inefficient. Limited applications of AltQ mostly rely on substantially altering the algorithm architecture in order to improve its performance. Although Adam appears to be a natural solution, its performance in AltQ has rarely been studied before. In this paper, we first provide a solid exploration on how well AltQ performs with Adam. We then take a further step to improve the implementation by adopting the technique of parameter restart. More specifically, the proposed algorithms are tested on a batch of Atari 2600 games and exhibit superior performance than the DQN learning method. The convergence rate of the slightly modified version of the proposed algorithms is characterized under the linear function approximation. To the best of our knowledge, this is the first theoretical study on the Adam-type algorithms in Q-learning.

## 1 INTRODUCTION

Q-learning (Watkins & Dayan, 1992) is one of the most important model-free reinforcement learning (RL) problems, which has received considerable research attention in recent years (Bertsekas & Tsitsiklis, 1996; Even-Dar & Mansour, 2003; Hasselt, 2010; Lu et al., 2018; Achiam et al., 2019). When the state-action space is large or continuous, parametric approximation of the Q-function is often necessary. One remarkable success of parametric Q-learning in practice is its combination with deep learning, known as the Deep Q-Network (DQN) learning (Mnih et al., 2013; 2015). It has been applied to various applications in computer games (Bhatti et al., 2016), traffic control (Arel et al., 2010), recommendation systems (Zheng et al., 2018; Zhao et al., 2018), chemistry research (Zhou et al., 2017), etc. Its on-policy continuous variant (Silver et al., 2014) has also led to great achievements in robotics locomotion (Lillicrap et al., 2016).

The DQN algorithm is performed in a nested-loop manner, where the outer loop follows an one-step update of the Q-function (via the empirical Bellman operator for Q-learning), and the inner loop takes a supervised learning process to fit the updated (i.e., target) Q-function with a neural network. In practice, the inner loop takes a sufficiently large number of iterations under certain optimizer (e.g. stochastic gradient descent (SGD) or Adam) to fit the neural network well to the target Q-function.

In contrast, a conventional Q-learning algorithm runs only one SGD step in each inner loop, in which case the overall Q-learning algorithm updates the Q-function and fits the target Q-function *alternatively* in each iteration. We refer to such a Q-learning algorithm with alternating updates as *Alternating Q*-learning (AltQ). Although significantly simpler in the update rule, AltQ is well known to be unstable and have weak performance (Mnih et al., 2016). This is in part due to the fact that the inner loop does not fit the target Q-function sufficiently well. To fix this issue, Mnih et al. (2016) proposed a new exploration strategy and asynchronous sampling schemes over parallel computing units (rather than the simple replay sampling in DQN) in order for the AltQ algorithm to achieve comparable or better performance than DQN. As another alternative, Knight & Lerner (2018) proposed a more involved natural gradient propagation for AltQ to improve the performance. All these schemes require more sophisticated designs or hardware support, which may place AltQ less advantageous compared to the popular DQN, even with their better performances. This motivates us to ask the following first question.

- *Q1: Can we design a simple and easy variant of the AltQ algorithm, which uses as simple setup as DQN and does not introduce extra computational burden and heuristics, but still achieves better and more stable performance than DQN?*

  In this paper, we provide an affirmative answer by introducing novel lightweight designs to AltQ based on Adam. Although Adam appears to be a natural tool, its performance in AltQ has rarely been studied yet. Thus, we first provide a solid exploration on how well AltQ performs with Adam (Kingma & Ba, 2014), where the algorithm is referred to as AltQ-Adam. We then take a further step to improve the implementation of AltQ-Adam by adopting the technique of parameter restart (i.e., restart the initial setting of Adam parameters every a few iterations), and refer to the new algorithm as AltQ-AdamR. This is the first time that restart is applied for improving the performance of RL algorithms although restart has been used for conventional optimization before.

  In a batch of 23 Atari 2600 games, our experiments show that both AltQ-Adam and AltQ-AdamR outperform the baseline performance of DQN by $50\%$ on average. Furthermore, AltQ-AdamR effectively reduces the performance variance and achieves a much more stable learning process. In our experiments for the linear quadratic regulator (LQR) problems, AltQ-AdamR converges even faster than the model-based value iteration (VI) solution. This is a rather surprising result given that the model-based VI has been treated as the performance upper bound for the Q-learning (including DQN) algorithms with target update (Lewis & Vrabie, 2009; Yang et al., 2019).

Regarding the theoretical analysis of AltQ algorithms, their convergence guarantee has been extensively studied (Melo et al., 2008; Chen et al., 2019b). More references are given in Section 1.1. However, all the existing studies focus on the AltQ algorithms that take a simple SGD step. Such theory is not applicable to the proposed AltQ-Adam and AltQ-AdamR that implement the Adam-type update. Thus, the second intriguing question we address here is described as follows.

- *Q2: Can we provide the convergence guarantee for AltQ-Adam and AltQ-AdamR or their slightly modified variants (if these two algorithms do not always converge by nature)?*

  It is well known in optimization that Adam does not always converge, and instead, a slightly modified variant AMSGrad proposed in Reddi et al. (2018) has been widely accepted as an alternative to justify the performance of Adam-type algorithms. Thus, our theoretical analysis here also focuses on such slightly modified variants AltQ-AMSGrad and AltQ-AMSGradR of the proposed algorithms. We show that under the linear function approximation (which is the structure that the current tools for analysis of Q-learning can handle), both AltQ-AMSGrad and AltQ-AMSGradR converge to the global optimal solution under standard assumptions for Q-learning. To the best of our knowledge, this is the first non-asymptotic convergence guarantee on Q-learning that incorporates Adam-type update and momentum restart. Furthermore, a slight adaptation of our proof provides the convergence rate for the AMSGrad for conventional strongly convex optimization which has not been studied before and can be of independent interest.

**Notations** We use $\|x\| := \|x\|_2 = \sqrt{x^T x}$ to denote the $\ell 2$ norm of a vector $x$, and use $\|x\|_\infty = \max_i |x_i|$ to denote the infinity norm. When $x, y$ are both vectors, $x/y, xy, x^2, \sqrt{x}$ are all calculated in the element-wise manner, which will be used in the update of Adam and AMSGrad. We denote $[n] = 1, 2, \ldots, n$, and $\lfloor x \rfloor \in \mathbb{Z}$ as the largest integer such that $\lfloor x \rfloor \leq x < \lfloor x \rfloor + 1$.

## 1.1 RELATED WORK

**Empirical performance of AltQ:** AltQ algorithms that strictly follow the alternating updates are rarely used in practice, particularly in comparison with the well-accepted DQN learning and its improved variants of dueling network structure (Wang et al., 2016), double Q-learning (Hasselt, 2010) and variance exploration and sampling schemes (Schaul et al., 2015). Mnih et al. (2016) proposed the asynchronous one-step Q-learning that is conceptually close to AltQ with competitive performance against DQN. However, the algorithm still relies on a slowly moving target network like DQN, and the multi-thread learning also complicates the computational setup. Lu et al. (2018) studied the problem of value overestimation and proposed the non-delusional Q-learning algorithm that employed the so-called pre-conditioned Q-networks, which is also computationally complex. Knight & Lerner (2018) proposed a natural gradient propagation for AltQ to improve the performance, where the gradient implementation is complex. In this paper, we propose two simple and computationally efficient schemes to improve the performance of AltQ.

**Theoretical analysis of AltQ:** Since proposed in Watkins & Dayan (1992), Q-learning has aroused great interest in theoretic analysis. The line of theoretic research of AltQ that are most relevant to our study lies in the Q-learning with function approximation. A large number of works study Q-learning with linear function approximation such as Bertsekas & Tsitsiklis (1996); Devraj & Meyn (2017); Zou et al. (2019); Chen et al. (2019b); Du et al. (2019), to name a few. More recently, convergence of AltQ with neural network parameterization was given in Cai et al. (2019), which exploits the linear structure of neural networks in the overparamterized regime for analysis. It is worth noting that all the existing analysis of AltQ with function approximation considers the simple SGD update, whereas our analysis in this paper focuses on the more involved Adam-type updates.

**Convergence analysis of Adam:** Adam was proposed in Kingma & Ba (2014) and has achieved a great success in training deep neural networks. Kingma & Ba (2014) and Reddi et al. (2018) provided regret bounds under the online convex optimization framework for Adam and AMSGrad, respectively. However, Tran et al. (2019) pointed out errors in the proofs of the previous two papers and corrected them. Recently, convergence analysis of Adam and AMSGrad in nonconvex optimization was provided in Zou et al. (2018); Zhou et al. (2018); Chen et al. (2019a); Phuong & Phong (2019), in which the Adam-type algorithms were guaranteed to converge to a stationary point. To the best of our knowledge, our study is the first convergence analysis of the Adam-type of algorithms for Q-learning.

## 2 PRELIMINARIES

We consider a Markov decision process with a considerably large or continuous state space $\mathcal{S} \subset \mathbb{R}^M$ and action space $\mathcal{A} \subset \mathbb{R}^N$ with a non-negative bounded reward function $R : \mathcal{S} \times \mathcal{A} \to [0, R_{\max}]$. We define $U(s) \subset \mathcal{A}$ as the admissible set of actions at state $s$, and $\pi : \mathcal{S} \to \mathcal{A}$ as a feasible stationary policy. We seek to solve a discrete-time sequential decision problem with $\gamma \in (0, 1)$ as follows:

$$\underset{\pi}{\text{maximize}} \quad J_\pi(s_0) = \mathbb{E}_P \left[ \sum_{t=0}^{\infty} \gamma^t R(s_t, \pi(s_t)) \right],$$

$$\text{subject to} \quad s_{t+1} \sim P(\cdot | s_t, a_t). \tag{1}$$

Let $J^\star(s) := J_{\pi^\star}(s)$ be the optimal value function when applying the optimal policy $\pi^\star$. The corresponding optimal Q-function can be defined as

$$Q^\star(s, a) := R(s, a) + \gamma \mathbb{E}_P J^\star(s'), \tag{2}$$

where $s' \sim P(\cdot | s, a)$ and we use the same notation hereafter when no confusion arises. In other words, $Q^\star(s, a)$ is the reward of an agent who starts from state $s$ and takes action $a$ at the first step and then follows the optimal policy $\pi^\star$ thereafter.

### 2.1 ALTQ ALGORITHM

This paper focuses on the Alternating Q-learning (AltQ) algorithm that uses a parametric function $\hat{Q}(s, a; \theta)$ to approximate the Q-function with a parameter $\theta$ of finite and relatively small dimension. The update rule of AltQ-learning is given by

$$T\hat{Q}(s, a; \theta_t) = R(s, a) + \gamma \max_{a' \in U(s')} \hat{Q}(s', a'; \theta_t); \tag{3}$$

$$\theta_{t+1} = \theta_t - \alpha_t \left( \hat{Q}_t(s, a; \theta_t) - T\hat{Q}(s, a; \theta_t) \right) \frac{\partial}{\partial \theta_t} \hat{Q}_t(s, a; \theta_t), \tag{4}$$

where $\alpha_t$ is the step size at time $t$. It is immediate from the equations that AltQ performs the update by taking one step temporal target update and one step parameter learning in an alternating fashion.

### 2.2 DQN ALGORITHM

As DQN is also included in this work for performance comparison. We recall the update of DQN in the following as reference. Differently from AltQ, DQN updates the parameters in a nested loop. Within the $t$-th inner loop, DQN first obtains the target Q-function as in Equation (5), and then uses a

neural network to fit the target Q-function by running $Y$ steps of a certain optimization algorithm as Equation (6). The update rule of DQN is given as follows.

$$T\hat{Q}(s, a; \theta_t^0) = R(s, a) + \gamma \max_{a' \in U(s')} \hat{Q}(s', a'; \theta_t^0), \tag{5}$$

$$\theta_t^Y = Optimizer(\theta_t^0, T\hat{Q}(s, a; \theta_t^0)), \tag{6}$$

where $Optimizer$ can be SGD or Adam for example, and Equation (6) is thus a supervised learning process with $T\hat{Q}(s, a; \theta_t^0))$ as the "supervisor". At the $t$-th outer loop, DQN performs the so-called target update as

$$\theta_{t+1}^0 = (1 - \tau)\theta_t^0 + \tau\theta_t^Y. \tag{7}$$

In practice, when one of the momentum-based optimizers is adopted for Equation (6), such as Adam, it is only initialized once at the beginning of the first inner loop. The historical gradient terms then accumulate throughout multiple inner loops with different targets. While this stabilizes the DQN training empirically, it is still lack of theoretical understanding on how the optimizer affects the training with various moving targets. As we will discuss in detail in Section 5, the analysis of AltQ with Adam can potentially shed light on such ambiguity and inspire future work for this matter.

Note that AltQ and DQN mainly differ at how the Q-function evolves after each step of sampling. A fair comparison between the algorithms should be made without introducing dramatic difference on gradient propagation (Knight & Lerner, 2018), policy structure, exploration and sampling strategies (Mnih et al., 2016). In practice, the vanilla AltQ is often slow in convergence and unstable with high variance. To improve the performance, we propose to incorporate Adam and restart schemes, which are easy to implement and yield improved performance than DQN.

## 3 Accelerated Alternating Q-learning Algorithms

In this section, we first describe how to incorporate Adam to the AltQ algorithm, and then introduce a novel implementation scheme to improve the performance of AltQ with Adam.

**AltQ with Adam-type update** We propose a new AltQ algorithm with Adam-type update (AltQ-Adam) as described in Algorithm 1. Its update is similar to the well-known Adam (Kingma & Ba, 2014). The iterations evolve by updating the exponentially decaying average of historical gradients ($m_t$) and squared historical gradients ($v_t$). The hyper-parameters $\beta_1, \beta_2$ are used to exponentially decrease the rate of the moving averages. The difference between Algorithm 1 and the standard Adam in supervised learning is that in AltQ, there is no fixed target to "supervise" the learning process. The target is always moving along with iteration $t$, leading to more noisy gradient estimations. The proposed algorithm sheds new light on the possibility of using Adam to deal with such unique challenge brought by RL.

**AltQ-Adam with momentum restart** We also introduce the restart technique to AltQ-Adam and propose AltQ-AdamR as Algorithm 2. Traditional momentum-based algorithms largely depend on the historical gradient direction. When part of the historical information is incorrect, the estimation error tends to accumulate. The restart technique can be employed to deal with this issue. One way to restart the momentum-based methods is to initialize the momentum at some restart iteration. That is, at restart iteration $r$, we reset $m_r, v_r$, i.e., $m_r = 0, v_r = 0$, which yields $\theta_{r+1} = \theta_r$. It is an intuitive implementation technique to adjust the trajectory from time to time, and can usually help mitigate the aforementioned problem while keeping fast convergence property. For the implementation, we execute the restart periodically with a period $r$. It turns out that the restart technique can significantly improve the numerical performance, which can be seen in Section 4.

## 4 Empirical Performance

We empirically evaluate the proposed algorithms in this section. The linear quadratic regulator (LQR) is a direct numerical demonstration of the convergence analysis under linear function approximation which will be discussed in the next section. Atari 2600 game (Bellemare et al., 2013; Brockman et al., 2016), a classic benchmark for DQN evaluations, is also used to show the effectiveness of the proposed algorithms for complicated tasks. In practice, we also make a small adjustment to the proposed algorithms. That is, we re-scale the loss term of $L(\theta_t) := \hat{Q}_t(s, a; \theta_t) - T\hat{Q}(s, a; \theta_t)$

---

**Algorithm 1** AltQ-Adam

---

1: **Input:** $\eta, \theta_1, \beta_1, \beta_2, \epsilon, \gamma, m_0 = 0, v_0 = 0.$
2: **for** $t = 1, 2, \ldots, K$ **do**
3:     $T\hat{Q}(s, a; \theta_t) = R(s, a) + \gamma \max_{a'} \hat{Q}(s', a'; \theta_t)$
4:     $g_t = \left(\hat{Q}_t(s, a; \theta_t) - T\hat{Q}(s, a; \theta_t)\right) \frac{\partial}{\partial \theta_t} \hat{Q}_t(s, a; \theta_t)$
5:     $m_t = (1 - \beta_1)m_{t-1} + \beta_1 g_t$
6:     $v_t = (1 - \beta_2)v_{t-1} + \beta_2 g_t^2$
7:     $\theta_{t+1} = \theta_t - \eta \frac{m_t}{\sqrt{v_t} + \epsilon}$
8: **end for**
9: **Output:** $\theta_K$

---

**Algorithm 2** AltQ-AdamR

---

1: **Input:** $\eta, \theta_1, \beta_1, \beta_2, \epsilon, \gamma, m_0 = 0, v_0 = 0, r.$
2: **for** $t = 1, 2, \ldots, K$ **do**
3:     **if** $\mod(t, r) = 0$ **then**
4:         $m_t = 0, v_t = 0$
5:     **end if**
6:     $T\hat{Q}(s, a; \theta_t) = R(s, a) + \gamma \max_{a'} \hat{Q}(s', a'; \theta_t)$
7:     $g_t = \left(\hat{Q}_t(s, a; \theta_t) - T\hat{Q}(s, a; \theta_t)\right) \frac{\partial}{\partial \theta_t} \hat{Q}_t(s, a; \theta_t)$
8:     $m_t = (1 - \beta_1)m_{t-1} + \beta_1 g_t$
9:     $v_t = (1 - \beta_2)v_{t-1} + \beta_2 g_t^2$
10:     $\theta_{t+1} = \theta_t - \eta \frac{m_t}{\sqrt{v_t} + \epsilon}$
11: **end for**
12: **Output:** $\theta_K$

---

in Equation (4) as $\tilde{L}(\theta_t) = \tilde{\tau}^2 L(\theta_t)$ with some scaling factor $\tilde{\tau} \in (0, 1]$, which is beneficial for stabilizing the learning process.

We find that in both experiments, AltQ-AdamR outperforms both AltQ-Adam and DQN in terms of convergence speed and variance reduction. Compared with DQN in the empirical experiments of Atari games, under the same hyper-parameter settings, AltQ-Adam and AltQ-AdamR improve the performance of DQN by 50% on average.

### 4.1 LINEAR QUADRATIC REGULATOR

We numerically validate the proposed algorithms through an infinite-horizon discrete-time LQR problem whose background can be found in Appendix A.1. A typical model-based solution (with known dynamics), known as the discrete-time algebraic Riccati equation (DARE), is adopted to derive the optimal policy $u_t^\star = -K^\star x_t$. The performance of the learning algorithm is then evaluated at each step of iterate $t$ with the Euclidean norm $\|K_t - K^\star\|$. The performance result for each method is averaged over 10 trials with different random seeds. All algorithms share the same set of random seeds and are initialized with the same $\theta_0$. The hyper-parameters of the learning settings are also consistent and further details are shown in Table 1. Note that for all the implementations, we also adopt the double Q-update (Hasselt, 2010) to help prevent over-estimations of the Q-value. The performance results are seen in Figure 1. Here we highlight main observations from the LQR experiments.

- **AltQ-AdamR outperforms DARE** In ideal cases where data sampling perfectly emulates the system dynamics and the target is accurately learned in each inner loop, DARE for LQR would become equivalent to the DQN-like update if the neural network is replaced with a parameterzied linear function. In practice, such ideal conditions are difficult to satisfy, and hence the actual Q-learning with target update is usually far slower (in terms of number of steps of target updates) than DARE. Note that AltQ-AdamR performs significantly well and even converges faster than DARE, and thus implies it is faster than the most well-performing Q-learning with target update.
- **AltQ-AdamR outperforms AltQ-Adam** Overall, under the same batch sampling scheme and restart period, AltQ-AdamR achieves a faster convergence and smaller variance than AltQ-Adam.

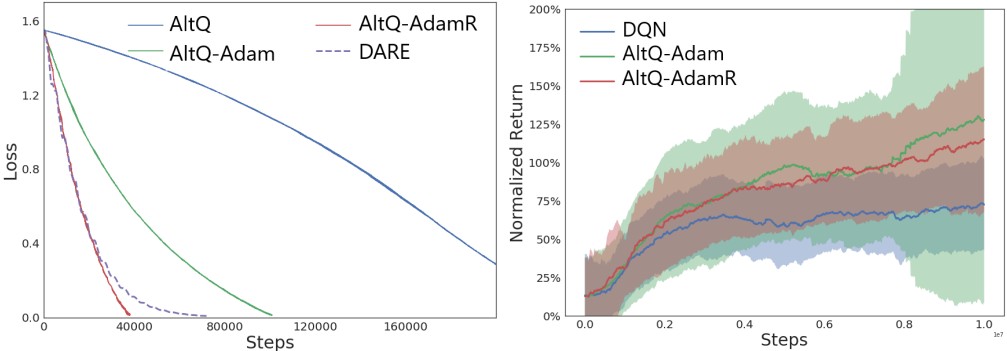

Figure 1: LQR experiments with performance evaluated in terms of policy loss $\|K_t - K^\star\|_2$.

Figure 2: Atari game experiment with performance normalized and averaged over 23 games.

Table 1: Hyper-parameters for LQR experiments

| Step size | $\tilde{\tau}$ | Adam $\beta_1$ | Adam $\beta_2$ | Restart period $r$ | Stop criterion | $\gamma$ |
|---|---|---|---|---|---|---|
| 0.0001 | 0.01 | 0.9 | 0.999 | 100 | $\|K_i - K^\star\|_2 \le 10^{-4}$ | 1 |

## 4.2 ATARI GAMES

We apply the proposed AltQ algorithms to more challenging tasks of deep convolutional neural network playing a group of Atari 2600 games. The particular DQN we train to compare against adopts the dueling network structure (Wang et al., 2016), double Q-learning setup (Van Hasselt et al., 2016), $\epsilon$-greedy exploration and experience replay (Mnih et al., 2013). Adam is also adopted, without momentum restart, as the optimizer for the inner-loop supervised learning process. AltQ-Adam and AltQ-AdamR are implemented using the identical setup of network construction, exploration and sampling strategies.

We test all the three algorithms with a batch of 23 Atari games. The choice of 10 million steps of iteration is a common setup for benchmark experiments with Atari games. Although this does not guarantee the best performance in comparison with more time-consuming training with 50 million steps or more, it is sufficient to illustrate different performances among the selected methods. The software infrastructure is based on the baseline implementation of OpenAI. Selections of the hyper-parameters are listed in Table 2. We summarize the results in Figure 2. The overall performance is illustrated by first normalizing the return of each method with respect to the results obtained from DQN, and then averaging the performance of all 23 games to obtain the mean return and standard deviation. Considering we use a smaller buffer size than common practice, DQN is not consistently showing improved return over all tested games. Therefore, the self-normalized average return of DQN in Figure 2 is not strictly increasing from 0 to 100%.

Overall, both AltQ-Adam and AltQ-AdamR achieve significant improvement in comparison with the DQN results. While AltQ-Adam is suffering from a higher variance, periodic restart (AltQ-AdamR) resolves the issue efficiently with an on-par performance on average and far smaller variance. Specifically, in terms of the maximum average return, AltQ-Adam and AltQ-AdamR perform no worse then DQN on 17 and 20 games respectively out of the 23 games being evaluated.

## 5 CONVERGENCE ANALYSIS

In this section, we characterize the convergence guarantee for the proposed AltQ-learning algorithms. Furthermore, like most of the related papers, we focus on convergence analysis under the linear approximation class. Understanding the analytical behavior in the linear case is an important stepping stone to understand general cases such as deep neural network. A linear approximation of the Q-function $\hat{Q}(s, a; \theta)$ can be written as

$$\hat{Q}(s, a; \theta) = \Phi(s, a)^T \theta, \tag{8}$$

where $\theta \in \mathbb{R}^d$, and $\Phi : \mathcal{S} \times \mathcal{A} \to \mathbb{R}^d$ is a vector function of size $d$, and the elements of $\Phi$ represent the nonlinear kernel (feature) functions.

Table 2: Hyper-parameters for Atari games experiments of DQN, AltQ-Adam and AltQ-AdamR

| Step size | Scale factor $\tilde{\tau}$ | Adam $\beta_1$ | Adam $\beta_2$ | Restart period $r$ | Buffer size |
|---|---|---|---|---|---|
| 0.0001 | 0.0001 | 0.9 | 0.999 | $10^4$ | $10^5$ |
| $\gamma$ | Batch size $B$ | Total training steps $K$ | Target update frequency (DQN only) | | |
| 0.99 | 32 | $10^7$ | $10^4$ | | |

### 5.1 MODIFICATION OF ALGORITHMS

Although Adam has obtained great success as an optimizer in deep learning, it is well known that Adam by nature is non-convergent even for simple convex loss functions (Reddi et al., 2018). Instead, a slightly modified version called AMSGrad (Reddi et al., 2018) is widely used to study the convergence property of the Adam-type algorithms. Compared with the update rule of Adam, AMSGrad makes the sequence $\hat{v}_{t,i}$ increasing along the time step $t$ for each entry $i \in [d]$. Here, we apply the update rule of AMSGrad to the AltQ algorithm and refer to such an algorithm as AltQ-AMSGrad. Algorithm 3 describes AltQ-AMSGrad in detail, where $\Pi_{\mathcal{D},\hat{V}_t^{1/4}}(\theta') = \min_{\theta \in \mathcal{D}} \left\| \hat{V}_t^{1/4}(\theta' - \theta) \right\|$. Correspondingly, we introduce AltQ-AMSGradR which applies the same update rule as Algorithm 3, but resets $m_t, \hat{v}_t$ with a period of $r$, i.e., $m_t = 0, \hat{v}_t = 0, \forall t = kr, k = 1, 2, \cdots$.

---

**Algorithm 3** AltQ-AMSGrad

1: **Input:** $\alpha, \lambda, \theta_1, \beta_1, \beta_2, m_0 = 0, \hat{v}_0 = 0$.
2: **for** $t = 1, 2, \ldots, T$ **do**
3: $\quad \alpha_t = \frac{\alpha}{\sqrt{t}}, \beta_{1t} = \beta_1 \lambda^t$
4: $\quad g_t = \left( \phi^T(s_t, a_t)\theta_t - r(s_t, a_t) - \max_{a'} \phi^T(s_{t+1}, a')\theta_t \right) \phi(s_t, a_t)$
5: $\quad m_t = (1 - \beta_{1t})m_{t-1} + \beta_{1t}g_t$
6: $\quad v_t = (1 - \beta_2)\hat{v}_{t-1} + \beta_2 g_t^2$
7: $\quad \hat{v}_t = \max(\hat{v}_{t-1}, v_t), \hat{V}_t = diag(\hat{v}_1, \ldots, \hat{v}_d)$
8: $\quad \theta_{t+1} = \Pi_{\mathcal{D},\hat{V}_t^{1/4}} \left( \theta_t - \alpha_t \hat{V}_t^{-\frac{1}{2}} m_t \right)$
9: **end for**
10: **Output:** $\frac{1}{T} \sum_{t=1}^T \theta_t$

---

### 5.2 MAIN RESULTS

Our theoretical analysis here focuses on the slight variants, AltQ-AMSGrad and AltQ-AMSGradR. Before stating the theorems, we first introduce some technical assumptions for our analysis.

**Assumption 1.** *At each iteration $t$, the noisy gradient is unbiased and uniformly bounded, i.e. $g_t = \bar{g}_t + \xi_t$ with $\mathbb{E}\xi_t = 0$ where $\bar{g}_t = \mathbb{E}[g_t]$, and $\|g_t\| < G_\infty, \forall t$. Thus $\|g_t\|_\infty < G_\infty$ and $\|g_t\|^2 < G_\infty^2$.*

**Assumption 2.** *(Chen et al., 2019b, Lemma 6.7) The equation $\bar{g}(\theta) = 0$ has a unique solution $\theta^\star$, which implies that there exists a $c > 0$, such that for any $\theta \in \mathbb{R}^d$ we have*

$$(\theta - \theta^\star)^T \bar{g}(\theta) \geq c \|\theta - \theta^\star\|^2. \tag{9}$$

**Assumption 3.** *The domain $\mathcal{D} \subset \mathbb{R}^d$ of approximation parameters is a ball originating at $\theta = 0$ with bounded diameter containing $\theta^\star$. That is, there exists $D_\infty$, such that $\|\theta_m - \theta_n\| < D_\infty, \forall \theta_m, \theta_n \in \mathcal{D}$, and $\theta^\star \in \mathcal{D}$.*

Assumption 1 is standard in the theoretical analysis of Adam-type algorithms (Chen et al., 2019a; Zhou et al., 2018). Under linear function approximation and given Assumption 3 and bounded $r(\cdot)$, Assumption 1 is almost equivalent to the assumption of bounded $\phi(\cdot)$ which is commonly taken in related RL work (Tsitsiklis & Van Roy, 1997; Bhandari et al., 2018). Assumption 2 has been proved as a key technical lemma in Chen et al. (2019b) under certain assumptions. Such an assumption appears to be the weakest in the existing studies of the theoretic guarantee for Q-learning with function approximation.

We next provide the convergence results of AltQ-AMSGrad and AltQ-AMSGradR under linear function approximation in the following two theorems.

**Theorem 1.** *(Convergence of AltQ-AMSGrad) Suppose $\alpha_t = \frac{\alpha}{\sqrt{t}}, \beta_{1t} = \beta_1 \lambda^t$ and $\delta = \beta_1/\beta_2$ with $\delta, \lambda \in (0,1)$ for $t = 1, 2, \ldots$ in Algorithm 3. Given Assumptions $1 \sim 3$, the output of AltQ-AMSGrad satisfies:*

$$\mathbb{E} \|\theta_{out} - \theta^\star\| \leq \frac{B_1}{T} + \frac{B_2 \sqrt{T}}{T} + \frac{B_3 \sqrt{1 + \log T}}{T} \sum_{i=1}^{d} \mathbb{E} \|g_{1:T,i}\|, \qquad (10)$$

*where $B_1 = \frac{G_\infty D_\infty^2}{2\alpha_2 c(1-\beta_1)} + \frac{\beta_1 G_\infty D_\infty^2}{2\alpha c(1-\beta_1)(1-\lambda)^2} + \|\theta_1 - \theta^\star\|^2, B_2 = \frac{dG_\infty D_\infty^2}{2\alpha c(1-\beta_1)}$, and $B_3 = \frac{\alpha(1+\beta_1)}{2c(1-\beta_1)^2(1-\delta)\sqrt{1-\beta_2}}$.*

In Theorem 1, $B_1, B_2, B_3$ in the bound in Equation (10) are constants and independent of time. Therefore, under the choice of the stepsize and hyper-parameters in Algorithm 3, AltQ-AMSGrad achieves a convergence rate of $\mathcal{O}\left(\frac{1}{\sqrt{T}}\right)$ when $\sum_{i=1}^{d} \|g_{1:T,i}\| << \sqrt{T}$ which is justified in Duchi et al. (2011).

**Remark 1.** *Our proof of convergence here has two major differences from that for AMSGrad in Reddi et al. (2018): (a) The two algorithms are quite different. AltQ-AMSGrad is a Q-learning algorithm alternatively finding the best policy, whereas AMSGrad is an optimizer for conventional optimization and does not have alternating nature. (b) Our analysis is on the convergence rate whereas Reddi et al. (2018) provides regret bound. In fact, a slight modification of our proof also provides the convergence rate of AMSGrad for conventional strongly convex optimization, which can be of independent interest. Moreover, our proof avoids the theoretical error in the proof in Reddi et al. (2018) pointed out by Tran et al. (2019).*

In the following theorem, we provide the convergence result for AltQ-AMSGradR.

**Theorem 2.** *(Convergence of AltQ-AMSGradR) Under the same condition of Theorem 1, the output of AltQ-AMSGradR satisfies:*

$$\mathbb{E} \|\theta_{out} - \theta^\star\| \leq \frac{B_1}{T} + \frac{B_2 \sqrt{1 + \log T}}{T} \sum_{i=1}^{d} \mathbb{E} \|g_{1:T,i}\| + \frac{B_3}{T} \left( \sqrt{T} + \sum_{k=1}^{\lfloor T/r \rfloor} \sqrt{kr - 1} \right)$$
$$+ \frac{1}{T} \sum_{k=0}^{\lfloor T/r \rfloor} \left( \frac{G_\infty D_\infty^2}{\alpha} \sqrt{kr + 2} + 4c(1-\beta_1) \mathbb{E} \|\theta_{kr} - \theta^\star\|^2 \right), \qquad (11)$$

*where $B_1 = \frac{\beta_1 D_\infty^2 G_\infty}{2\alpha c(1-\beta_1)(1-\lambda)^2}, B_2 = \frac{\alpha(1+\beta_1)}{2c(1-\beta_1)^2(1-\delta)\sqrt{1-\beta_2}}$, and $B_3 = \frac{dG_\infty D_\infty^2}{2\alpha c(1-\beta_1)}$.*

Theorem 2 indicates that for AltQ-AMSGradR to enjoy a convergence rate of $\mathcal{O}\left(\frac{1}{\sqrt{T}}\right)$ the restart period $r$ needs to be sufficiently large and $\sum_{i=1}^{d} \|g_{1:T,i}\| << \sqrt{T}$. In practice as demonstrated by the experiments in Section 4, AltQ-AMSGradR typically performs well, not necessarily under the theoretical conditions.

## 6 CONCLUSION

We propose two types of the accelerated AltQ algorithms, and demonstrate their superior performance over the state-of-the-art through a linear quadratic regulator problem and a batch of 23 Atari 2600 games.

Notably, Adam is not the only scheme in the practice for general optimization. Heavy ball (Ghadimi et al., 2015) and Nesterov (Nesterov, 2013) are also popular momentum-based methods. When adopting such methods in AltQ-learning for RL problems, however, we tend to observe a less stable learning process than AltQ-Adam. This is partially caused by the fact that they optimize over a shorter historical horizon of updates than Adam. Furthermore, the restart scheme provides somewhat remarkable performance in our study. It is thus of considerable future interest to further investigate the potential of such a scheme. One possible direction is to develop an adaptive restart mechanism with changing period determined by an appropriately defined signal of restart. This will potentially relieve the effort in hyper-parameter tuning of finding a good fixed period.

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

# Supplementary Materials

## A FURTHER DETAILS AND RESULTS ON EXPERIMENTS

We discuss more details on the experiment setup and provide further results that are not included in Section 4.

### A.1 LINEAR QUADRATIC REGULATOR

The linear quadratic regulator (LQR) problem is of great interest for control community where Lewis et al. applies PQL to both discrete-time problems (Lewis & Vrabie, 2009; Lewis & Vamvoudakis, 2011) and continuous-time problems (Vamvoudakis, 2017; Vrabie et al., 2009).

We empirically validate the proposed algorithms through an infinite-horizon discrete-time LQR problem defined as

$$\underset{\pi}{\text{minimize}} \qquad J = \sum_{t=0}^{\infty} \left( x_t^T Q x_t + u_t^T R u_t + 2 x_t^T N u_t \right),$$
$$\text{subject to} \qquad x_{t+1} = A x_t + B u_t,$$

where $u_t = \pi(x_t)$.

A typical model-based solution (with known $A$ and $B$) considers the problem backwards in time and iterates a dynamic equation known as the discrete-time algebraic Riccati equation (DARE):

$$P = A^T P A - (A^T P B + N)(R + B^T P B)^{-1}(B^T P A + N^T) + Q, \qquad (12)$$

with the cost-to-go $P$ being positive definite. The optimal policy satisfies $u_t^\star = -K^\star x_t$ with

$$K^\star = (R + B^T P B)^{-1}(N^T + B^T P A). \qquad (13)$$

For experiments, we parameterize a quadratic Q-function with a matrix parameter $H$ in the form of

$$Q(x, u; H) = \begin{bmatrix} x \\ u \end{bmatrix}^T \begin{bmatrix} H_{xx} & H_{xu} \\ H_{ux} & H_{uu} \end{bmatrix} \begin{bmatrix} x \\ u \end{bmatrix}. \qquad (14)$$

The corresponding linear policy satisfies $u = -Kx$, and $K = H_{uu}^{-1} H_{ux}$. The performance of the learning algorithm is then evaluated at each step of iterate $i$ with the Euclidean norm $\|K_i - K^\star\|_2$.

### A.2 ATARI GAMES

We list detailed experiments of the 23 Atari games evaluated with the proposed algorithms in Figure 3. All experiments are executed with the same set of two random seeds. Each task takes about 20-hour of wall-clock time on a GPU instance. All three methods being evaluated share similar training time. AltQ-Adam and AltQ-AdamR can be further accelerated in practice with a more memory-efficient implementation considering the target network is not required. We keep our implementation of proposed algorithms consistent with the DQN we are comparing against. Other techniques that are not included in this experiment are also compatible with AltQ-Adam and AltQ-AdamR, such like asynchronous exploration (Mnih et al., 2013) and training with decorrelated loss (Mavrin et al., 2019).

Overall, AltQ-Adam significantly increases the performance by over $100\%$ in some of the tasks including Asterix, BeamRider, Enduro, Gopher, etc. However, it also illustrates certain instability with complete failure on Amidar and Assault. This is mostly caused by the sampling where we are using a relevantly small buffer size with $10\%$ of the common configured size in Atari games with experience replay. Notice that those failures tend to appear when the $\epsilon$-greedy exploration has evolved to a certain level where the immediate policy is effectively contributing to the accumulated experience. This potentially amplifies the biased exploration that essentially leads to the observed phenomenon.

Interstingly, AltQ-AdamR that incorporates the restart scheme resolves the problem of high variance of average return brought by AltQ-Adam and provides a more consistent performance across the task

domain. This implies that momentum restart effectively corrects the accumulated error and stabilizes the training process.

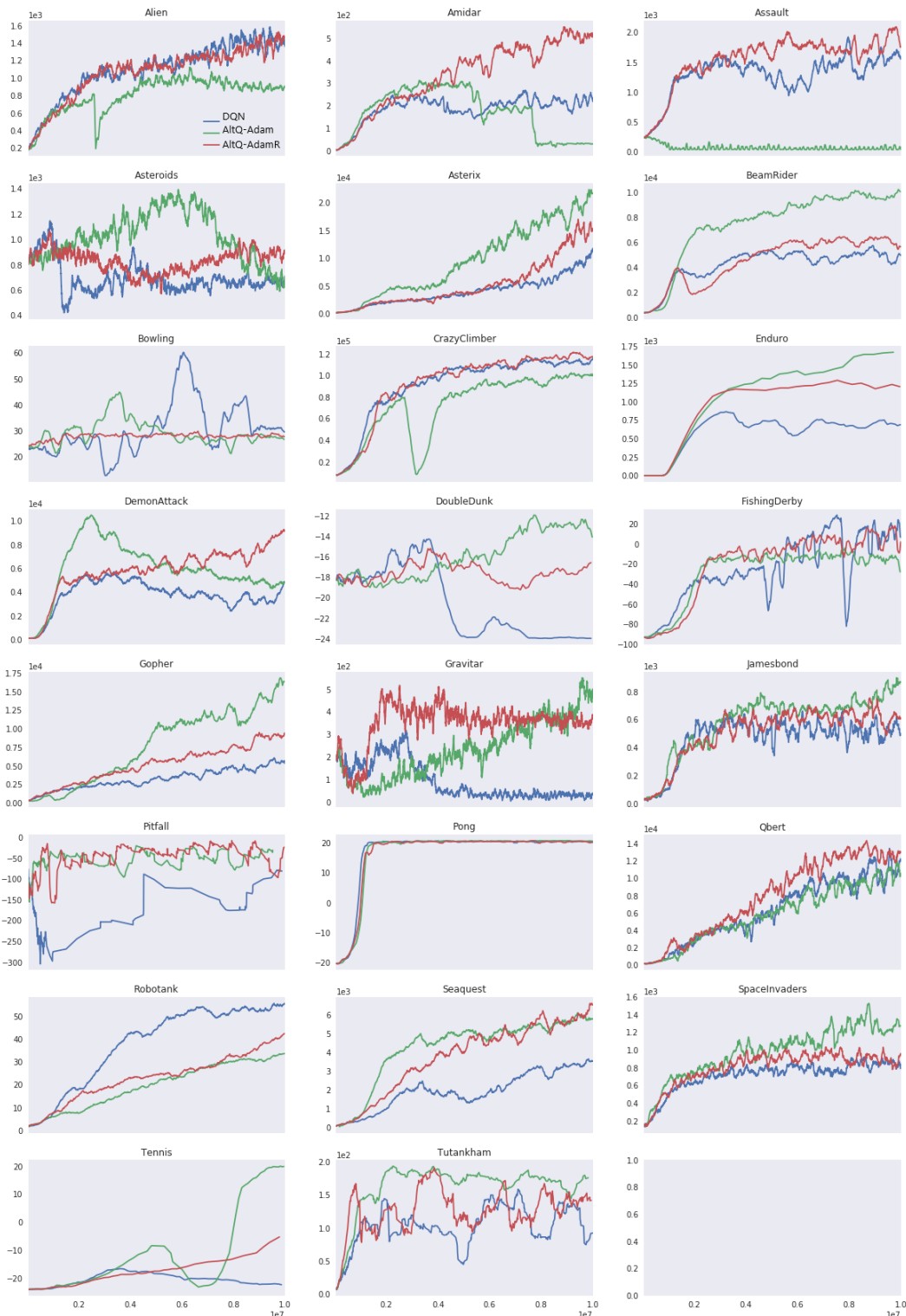

Figure 3: Experiment results of 23 Atari games with DQN, AltQ-Adam and AltQ-AdamR

| Task | DQN | AltQ-Adam | AltQ-AdamR |
|------|-----|-----------|------------|
| Alien | 1529 | 1125 | **1587** |
| Amidar | 269 | 313 | **551** |
| Assault | 1925 | 260 | **2097** |
| Asteroids | 1147 | **1394** | 1069 |
| Asterix | 11794 | **22413** | 17064 |
| BeamRider | 5728 | **10210** | 6458 |
| Bowling | **60** | 45 | 30 |
| CrazyClimber | 116422 | 102731 | **121770** |
| Enduro | 866 | **1671** | 1291 |
| DemonAttack | 5729 | **10485** | 9273 |
| DoubleDunk | -14 | **-12** | -15 |
| FishingDerby | **29.01** | -4 | 19 |
| Gopher | 6066 | **16863** | 9508 |
| Gravitar | 316 | **551** | 518 |
| Jamesbond | 663 | **899** | 756 |
| Pitfall | -76 | -20 | **-7** |
| Pong | 20.68 | **20.79** | 20.74 |
| Qbert | 13453 | 12487 | **14352** |
| Robotank | **56** | 34 | 42 |
| Seaquest | 3652 | 6121 | **6624** |
| Spaceinvaders | 923 | **1528** | 1036 |
| Tennis | -17 | **20** | -5 |
| Tutankham | 159 | **194** | 191 |

Table 3: Best empirical return of 23 Atari games with DQN, AltQ-Adam and AltQ-AdamR

## B  PROOF OF THEOREM 1

Different from the regret bound for AMSGrad obtained in Reddi et al. (2018), our analysis is on the convergence rate. In fact, a slight modification of our proof also provides the convergence rate for AMSGrad for conventional strongly convex optimization, which can be of independent interest. Moreover, our proof avoids the theoretical error in the proof in Reddi et al. (2018) pointed out by (Tran et al., 2019). Before proving the theorems, we first provide some useful lemmas.

**Lemma 1.** *(Zhou et al., 2018, Lemma A.1) Let $\{g_t, m_t, \hat{v}_t\}$ for $t = 1, 2, \ldots$ be sequences generated by Algorithm 3 and $\bar{g}_t = \mathbb{E}[g_t]$. Under Assumption 1, $\|\bar{g}_t\| \leq G_\infty, \|m_t\| \leq G_\infty, \|\hat{v}_t\| \leq G_\infty^2$.*

**Lemma 2.** *(Reddi et al., 2018, Lemma 2) Let $\{m_t, \hat{V}_t\}$ for $t = 1, 2, \ldots$ be sequences generated by Algorithm 3. Given $\alpha_t, \beta_{1t}, \beta_2$ as specified in Theorem 1, we have*

$$
\begin{aligned}
\sum_{t=1}^{T} \alpha_t \left\| \hat{V}_t^{-\frac{1}{2}} m_t \right\|^2 &\leq \frac{\alpha}{(1-\beta_1)(1-\delta)\sqrt{1-\beta_2}} \sum_{i=1}^{d} \|g_{1:T,i}\| \sqrt{\sum_{t=1}^{T} \frac{1}{t}} \\
&\leq \frac{\alpha\sqrt{1 + \log T}}{(1-\beta_1)(1-\delta)\sqrt{1-\beta_2}} \sum_{i=1}^{d} \|g_{1:T,i}\|.
\end{aligned}
$$
(15)

**Lemma 3.** *Let $\alpha_t = \frac{\alpha}{\sqrt{t}}$ and $\beta_{1t} = \beta_1 \lambda^t$ for $t = 1, 2, \ldots$. Then*

$$
\sum_{t=1}^{T} \frac{\beta_{1t}}{\alpha_t} \leq \frac{\beta_1}{\alpha(1-\lambda)^2}.
$$
(16)

*Proof.* The proof is based on taking the standard sum of geometric sequences.

$$
\sum_{t=1}^{T} \frac{\beta_{1t}}{\alpha_t} = \sum_{t=1}^{T} \frac{\beta_{1t}\sqrt{t}}{\alpha} \leq \sum_{t=1}^{T} \frac{\beta_1 \lambda^{t-1} t}{\alpha} = \frac{\beta_1}{\alpha} \left( \frac{1}{(1-\lambda)} \sum_{t=1}^{T} \lambda^{t-1} - T\lambda^T \right) \leq \frac{\beta_1}{\alpha(1-\lambda)^2}. \quad (17)
$$

$\square$

With the lemmas above, we are ready to prove Theorem 1. Observe that

$$\theta_{t+1} = \Pi_{\mathcal{D}, \hat{V}_t^{1/4}} \left( \theta_t - \alpha_t \hat{V}_t^{-\frac{1}{2}} m_t \right) = \min_{\theta \in \mathcal{D}} \left\| \hat{V}_t^{1/4} \left( \theta_t - \alpha_t \hat{V}_t^{-\frac{1}{2}} m_t - \theta \right) \right\|.$$

Clearly $\Pi_{\mathcal{D}, \hat{V}_t^{1/4}}(\theta^\star) = \theta^\star$ due to Assumption 3. We start from the update of $\theta_t$ when $t \geq 2$.

$$\left\| \hat{V}_t^{1/4} (\theta_{t+1} - \theta^\star) \right\|^2$$

$$= \left\| \Pi_{\mathcal{D}, \hat{V}_t^{1/4}} \hat{V}_t^{1/4} \left( \theta_t - \theta^\star - \alpha_t \hat{V}_t^{-\frac{1}{2}} m_t \right) \right\|^2$$

$$\leq \left\| \hat{V}_t^{1/4} \left( \theta_t - \theta^\star - \alpha_t \hat{V}_t^{-\frac{1}{2}} m_t \right) \right\|^2$$

$$= \left\| \hat{V}_t^{1/4} (\theta_t - \theta^\star) \right\|^2 + \left\| \alpha_t \hat{V}_t^{-1/4} m_t \right\|^2 - 2\alpha_t (\theta_t - \theta^\star)^T m_t$$

$$= \left\| \hat{V}_t^{1/4} (\theta_t - \theta^\star) \right\|^2 + \left\| \alpha_t \hat{V}_t^{-1/4} m_t \right\|^2 - 2\alpha_t (\theta_t - \theta^\star)^T (\beta_{1t} m_{t-1} + (1 - \beta_{1t}) g_t)$$

$$\overset{(i)}{\leq} \left\| \hat{V}_t^{1/4} (\theta_t - \theta^\star) \right\|^2 + \left\| \alpha_t \hat{V}_t^{-1/4} m_t \right\|^2 + \alpha_t \beta_{1t} \left( \frac{1}{\alpha_t} \left\| \hat{V}_t^{1/4} (\theta_t - \theta^\star) \right\|^2 + \alpha_t \left\| \hat{V}_t^{-1/4} m_{t-1} \right\|^2 \right)$$

$$- 2\alpha_t (1 - \beta_{1t}) (\theta_t - \theta^\star)^T g_t$$

$$\overset{(ii)}{\leq} \left\| \hat{V}_t^{1/4} (\theta_t - \theta^\star) \right\|^2 + \left\| \alpha_t \hat{V}_t^{-1/4} m_t \right\|^2 + \beta_{1t} \left\| \hat{V}_t^{1/4} (\theta_t - \theta^\star) \right\|^2 + \alpha_t^2 \beta_{1t} \left\| \hat{V}_{t-1}^{-1/4} m_{t-1} \right\|^2$$

$$- 2\alpha_t (1 - \beta_{1t}) (\theta_t - \theta^\star)^T g_t,$$

where (i) follows from the Cauchy-Schwarz inequality, and (ii) holds because $\hat{v}_{t+1,i} \geq \hat{v}_{t,i}, \forall t, \forall i$. Next, we take the expectation over all samples used up to time step $t$ on both sides, which still preserves the inequality. Since we consider i.i.d. sampling case, by letting $\mathcal{F}_t$ be the filtration of all the sampling up to time $t$, we have

$$\mathbb{E}\left[ (\theta_t - \theta^\star)^T g_t \right] = \mathbb{E}\left[ \mathbb{E}\left[ (\theta_t - \theta^\star)^T g_t \right] | \mathcal{F}_{t-1} \right] = \mathbb{E}\left[ (\theta_t - \theta^\star)^T \bar{g}_t \right]. \tag{18}$$

Thus we have

$$\mathbb{E}\left\| \hat{V}_t^{1/4} (\theta_{t+1} - \theta^\star) \right\|^2$$

$$\leq \mathbb{E}\left\| \hat{V}_t^{1/4} (\theta_t - \theta^\star) \right\|^2 + \alpha_t^2 \mathbb{E}\left\| \hat{V}_t^{-1/4} m_t \right\|^2 + \beta_{1t} \mathbb{E}\left\| \hat{V}_t^{1/4} (\theta_t - \theta^\star) \right\|^2 + \alpha_t^2 \beta_{1t} \mathbb{E}\left\| \hat{V}_{t-1}^{-1/4} m_{t-1} \right\|^2$$

$$- 2\alpha_t (1 - \beta_{1t}) \mathbb{E}\left[ (\theta_t - \theta^\star)^T g_t \right]$$

$$\overset{(i)}{=} \mathbb{E}\left\| \hat{V}_t^{1/4} (\theta_t - \theta^\star) \right\|^2 + \alpha_t^2 \mathbb{E}\left\| \hat{V}_t^{-1/4} m_t \right\|^2 + \beta_{1t} \mathbb{E}\left\| \hat{V}_t^{1/4} (\theta_t - \theta^\star) \right\|^2 + \alpha_t^2 \beta_{1t} \mathbb{E}\left\| \hat{V}_{t-1}^{-1/4} m_{t-1} \right\|^2$$

$$- 2\alpha_t (1 - \beta_{1t}) \mathbb{E}\left[ (\theta_t - \theta^\star)^T \bar{g}_t \right]$$

$$\overset{(ii)}{\leq} \mathbb{E}\left\| \hat{V}_t^{1/4} (\theta_t - \theta^\star) \right\|^2 + \alpha_t^2 \mathbb{E}\left\| \hat{V}_t^{-1/4} m_t \right\|^2 + \beta_{1t} \mathbb{E}\left\| \hat{V}_t^{1/4} (\theta_t - \theta^\star) \right\|^2 + \alpha_t^2 \beta_{1t} \mathbb{E}\left\| \hat{V}_{t-1}^{-1/4} m_{t-1} \right\|^2$$

$$- 2\alpha_t c (1 - \beta_{1t}) \mathbb{E}\|\theta_t - \theta^\star\|^2$$

$$\overset{(iii)}{\leq} \mathbb{E}\left\| \hat{V}_t^{1/4} (\theta_t - \theta^\star) \right\|^2 + \alpha_t^2 \mathbb{E}\left\| \hat{V}_t^{-1/4} m_t \right\|^2 + \beta_{1t} \mathbb{E}\left\| \hat{V}_t^{1/4} (\theta_t - \theta^\star) \right\|^2 + \alpha_t^2 \beta_1 \mathbb{E}\left\| \hat{V}_{t-1}^{-1/4} m_{t-1} \right\|^2$$

$$- 2\alpha_t c (1 - \beta_1) \mathbb{E}\|\theta_t - \theta^\star\|^2$$

$$\overset{(iv)}{\leq} \mathbb{E}\left\| \hat{V}_t^{1/4} (\theta_t - \theta^\star) \right\|^2 + \alpha_t^2 \mathbb{E}\left\| \hat{V}_t^{-1/4} m_t \right\|^2 + G_\infty D_\infty^2 \beta_{1t} + \alpha_t^2 \beta_1 \mathbb{E}\left\| \hat{V}_{t-1}^{-1/4} m_{t-1} \right\|^2$$

$$- 2\alpha_t c (1 - \beta_1) \mathbb{E}\|\theta_t - \theta^\star\|^2,$$

where (i) follows from Equation (18), (ii) follows due to Assumption 2 and $1 - \beta_{1t} > 0$, (iii) follows from $\beta_{1t} < \beta_1 < 1$ and $\mathbb{E}\|\theta_t - \theta^\star\|^2 > 0$, and (iv) follows from $\left\| \hat{V}_t^{1/4} (\theta_t - \theta^\star) \right\|^2 \leq \left\| \hat{V}_t^{1/4} \right\|_2^2 \|\theta_t - \theta^\star\|^2 \leq G_\infty D_\infty^2$ by Lemma 1 and Assumption 3. We note that (iii) is the key step to

avoid the error in the proof in Reddi et al. (2018), where we can directly bound $1 - \beta_{1t}$, which is impossible in Reddi et al. (2018). By rearranging the terms in the above inequality and taking the summation over time steps, we have

$$2c(1 - \beta_1) \sum_{t=2}^{T} \mathbb{E} \|\theta_t - \theta^\star\|^2$$

$$\leq \sum_{t=2}^{T} \frac{1}{\alpha_t} \left( \mathbb{E} \left\| \hat{V}_t^{1/4}(\theta_t - \theta^\star) \right\|^2 - \mathbb{E} \left\| \hat{V}_t^{1/4}(\theta_{t+1} - \theta^\star) \right\|^2 \right) + \sum_{t=2}^{T} \frac{\beta_{1t} G_\infty D_\infty^2}{\alpha_t}$$

$$+ \sum_{t=2}^{T} \alpha_t \mathbb{E} \left\| \hat{V}_t^{-1/4} m_t \right\|^2 + \sum_{t=2}^{T} \alpha_t \beta_1 \mathbb{E} \left\| \hat{V}_{t-1}^{-1/4} m_{t-1} \right\|^2$$

$$\overset{(i)}{\leq} \sum_{t=2}^{T} \frac{1}{\alpha_t} \left( \mathbb{E} \left\| \hat{V}_t^{1/4}(\theta_t - \theta^\star) \right\|^2 - \mathbb{E} \left\| \hat{V}_t^{1/4}(\theta_{t+1} - \theta^\star) \right\|^2 \right) + \sum_{t=2}^{T} \frac{\beta_{1t} G_\infty D_\infty^2}{\alpha_t}$$

$$+ \sum_{t=2}^{T} \alpha_t \mathbb{E} \left\| \hat{V}_t^{-1/4} m_t \right\|^2 + \sum_{t=2}^{T} \alpha_{t-1} \beta_1 \mathbb{E} \left\| \hat{V}_{t-1}^{-1/4} m_{t-1} \right\|^2$$

$$\leq \sum_{t=2}^{T} \frac{1}{\alpha_t} \left( \mathbb{E} \left\| \hat{V}_t^{1/4}(\theta_t - \theta^\star) \right\|^2 - \mathbb{E} \left\| \hat{V}_t^{1/4}(\theta_{t+1} - \theta^\star) \right\|^2 \right) + \sum_{t=2}^{T} \frac{\beta_{1t} G_\infty D_\infty^2}{\alpha_t}$$

$$+ (1 + \beta_1) \sum_{t=1}^{T} \alpha_t \mathbb{E} \left\| \hat{V}_t^{-1/4} m_t \right\|^2 ,$$

where (i) follows from $\alpha_t < \alpha_{t-1}$. With further adjustment of the first term in the right hand side of the last inequality, we can then bound the sum as

$$2c(1 - \beta_1) \sum_{t=2}^{T} \mathbb{E} \|\theta_t - \theta^\star\|^2$$

$$\leq \sum_{t=2}^{T} \frac{1}{\alpha_t} \mathbb{E} \left( \left\| \hat{V}_t^{1/4}(\theta_t - \theta^\star) \right\|^2 - \left\| \hat{V}_t^{1/4}(\theta_{t+1} - \theta^\star) \right\|^2 \right) + \sum_{t=2}^{T} \frac{\beta_{1t} G_\infty D_\infty^2}{\alpha_t}$$

$$+ (1 + \beta_1) \sum_{t=1}^{T} \alpha_t \mathbb{E} \left\| \hat{V}_t^{-1/4} m_t \right\|^2$$

$$= \frac{\mathbb{E} \left\| \hat{V}_2^{1/4}(\theta_2 - \theta^\star) \right\|^2}{\alpha_2} + \sum_{t=3}^{T} \mathbb{E} \left( \frac{\left\| \hat{V}_t^{1/4}(\theta_t - \theta^\star) \right\|^2}{\alpha_t} - \frac{\left\| \hat{V}_{t-1}^{1/4}(\theta_t - \theta^\star) \right\|^2}{\alpha_{t-1}} \right)$$

$$- \frac{\mathbb{E} \left\| \hat{V}_T^{1/4}(\theta_{T+1} - \theta^\star) \right\|^2}{\alpha_T} + \sum_{t=2}^{T} \frac{\beta_{1t} G_\infty D_\infty^2}{\alpha_t} + (1 + \beta_1) \sum_{t=1}^{T} \alpha_t \mathbb{E} \left\| \hat{V}_t^{-1/4} m_t \right\|^2$$

$$= \frac{\mathbb{E} \left\| \hat{V}_2^{1/4}(\theta_2 - \theta^\star) \right\|^2}{\alpha_2} + \sum_{t=3}^{T} \mathbb{E} \left( \frac{\sum_{i=1}^{d} \hat{v}_{t,i}^{1/2}(\theta_{t,i} - \theta_i^\star)^2}{\alpha_t} - \frac{\sum_{i=1}^{d} \hat{v}_{t-1,i}^{1/2}(\theta_{t,i} - \theta_i^\star)^2}{\alpha_{t-1}} \right)$$

$$- \frac{\mathbb{E} \left\| \hat{V}_T^{1/4}(\theta_{T+1} - \theta^\star) \right\|^2}{\alpha_T} + \sum_{t=2}^{T} \frac{\beta_{1t} G_\infty D_\infty^2}{\alpha_t} + (1 + \beta_1) \sum_{t=1}^{T} \alpha_t \mathbb{E} \left\| \hat{V}_t^{-1/4} m_t \right\|^2 .$$

$$= \frac{\mathbb{E} \left\| \hat{V}_2^{1/4}(\theta_2 - \theta^\star) \right\|^2}{\alpha_2} + \sum_{t=3}^{T} \sum_{i=1}^{d} \mathbb{E}(\theta_{t,i} - \theta_i^\star)^2 \left( \frac{\hat{v}_{t,i}^{1/2}}{\alpha_t} - \frac{\hat{v}_{t-1,i}^{1/2}}{\alpha_{t-1}} \right)$$

$$- \frac{\mathbb{E} \left\| \hat{V}_T^{1/4}(\theta_{T+1} - \theta^\star) \right\|^2}{\alpha_T} + \sum_{t=2}^{T} \frac{\beta_{1t} G_\infty D_\infty^2}{\alpha_t} + (1 + \beta_1) \sum_{t=1}^{T} \alpha_t \mathbb{E} \left\| \hat{V}_t^{-1/4} m_t \right\|^2 .$$

So far we just rearrange the terms in the series sum. Next, we are ready to obtain the upper bound.

$$2c(1 - \beta_1) \sum_{t=2}^{T} \mathbb{E} \left\| \theta_t - \theta^\star \right\|^2$$

$$\overset{(i)}{\leq} \frac{\mathbb{E} \left\| \hat{V}_2^{1/4} (\theta_2 - \theta^\star) \right\|^2}{\alpha_2} + D_\infty^2 \sum_{t=3}^{T} \sum_{i=1}^{d} \mathbb{E} \left( \frac{\hat{v}_{t,i}^{1/2}}{\alpha_t} - \frac{\hat{v}_{t-1,i}^{1/2}}{\alpha_{t-1}} \right)$$

$$- \frac{\mathbb{E} \left\| \hat{V}_T^{1/4} (\theta_{T+1} - \theta^\star) \right\|^2}{\alpha_T} + \sum_{t=2}^{T} \frac{\beta_{1t} G_\infty D_\infty^2}{\alpha_t} + (1 + \beta_1) \sum_{t=1}^{T} \alpha_t \mathbb{E} \left\| \hat{V}_t^{-1/4} m_t \right\|^2$$

$$\leq \frac{\mathbb{E} \left\| \hat{V}_2^{1/4} (\theta_2 - \theta^\star) \right\|^2}{\alpha_2} + D_\infty^2 \sum_{i=1}^{d} \mathbb{E} \frac{\hat{v}_{T,i}^{1/2}}{\alpha_T} + \sum_{t=2}^{T} \frac{\beta_{1t} G_\infty D_\infty^2}{\alpha_t} + (1 + \beta_1) \sum_{t=1}^{T} \alpha_t \mathbb{E} \left\| \hat{V}_t^{-1/4} m_t \right\|^2$$

$$\overset{(ii)}{\leq} \frac{G_\infty D_\infty^2}{\alpha_2} + \frac{d G_\infty D_\infty^2 \sqrt{T}}{\alpha} + \frac{\beta_1 G_\infty D_\infty^2}{\alpha (1 - \lambda)^2} + \frac{\alpha (1 + \beta_1) \sqrt{1 + \log T}}{(1 - \beta_1)(1 - \delta)\sqrt{1 - \beta_2}} \sum_{i=1}^{d} \mathbb{E} \left\| g_{1:T,i} \right\|,$$

$$(19)$$

where (i) follows from Assumption 3 and because $\frac{\hat{v}_{t,i}^{1/2}}{\alpha_t} > \frac{\hat{v}_{t-1,i}^{1/2}}{\alpha_{t-1}}$, and (ii) follows from Lemmas 1 - 3. Finally, applying the Jensen's inequality yields

$$\mathbb{E} \left\| \theta_{out} - \theta^\star \right\|^2 \leq \frac{1}{T} \sum_{t=1}^{T} \mathbb{E} \left\| \theta_t - \theta^\star \right\|^2. \tag{20}$$

We conclude our proof by further applying the bound in Equation (19) to Equation (20).

## C   PROOF OF THEOREM 2

To prove the convergence for AltQ-AMSGradR, the major technical development beyond the proof of Theorem 1 lies in dealing with the parameter restart. More specifically, the moment approximation terms are reset every $r$ steps, i.e., $m_{kr} = \hat{v}_{kr} = 0$ for $k = 1, 2, \ldots$, which implies $\theta_{kr+1} = \theta_{kr}$ for $k = 1, 2, \ldots$. For technical convenience, we define $\theta_0 = \theta_1$. Using the arguments similar to Equation (19), in a time window that does not contain a restart (i.e. $kr \le S \le (k+1)r - 1$) we have

$$2c(1 - \beta_1) \sum_{t=kr}^{S} \mathbb{E} \|\theta_t - \theta^\star\|^2$$

$$\stackrel{(i)}{\le} \frac{G_\infty D_\infty^2}{\alpha_{kr+2}} + \frac{dG_\infty D_\infty^2 \sqrt{S}}{\alpha} + \frac{\alpha(1 + \beta_1)}{(1 - \beta_1)(1 - \delta)\sqrt{1 - \beta_2}} \sum_{i=1}^{d} \mathbb{E} \|g_{kr+1:S,i}\| \sqrt{\sum_{t=kr+1}^{S} \frac{1}{t}}$$

$$+ G_\infty D_\infty^2 \sum_{t=kr+2}^{S} \frac{\beta_{1t}}{\alpha_t} + 2c(1 - \beta_1) \left( \mathbb{E} \|\theta_{kr+1} - \theta^\star\|^2 + \mathbb{E} \|\theta_{kr} - \theta^\star\|^2 \right)$$

$$\stackrel{(ii)}{=} \frac{G_\infty D_\infty^2 \sqrt{kr + 2}}{\alpha} + \frac{dG_\infty D_\infty^2 \sqrt{S}}{\alpha} + \frac{\alpha(1 + \beta_1)}{(1 - \beta_1)(1 - \delta)\sqrt{1 - \beta_2}} \sum_{i=1}^{d} \mathbb{E} \|g_{kr+1:S,i}\| \sqrt{\sum_{t=kr+1}^{S} \frac{1}{t}}$$

$$+ G_\infty D_\infty^2 \sum_{t=kr+2}^{S} \frac{\beta_{1t}}{\alpha_t} + 4c(1 - \beta_1)\mathbb{E} \|\theta_{kr} - \theta^\star\|^2,$$

where (i) follows from Equation (19) and (ii) follows from $\theta_{kr+1} = \theta_{kr}$ due to the definition of restart. Then we take the summation over the total time steps and obtain

$$2c(1 - \beta_1) \sum_{t=1}^{T} \mathbb{E} \|\theta_t - \theta^\star\|^2$$

$$= 2c(1 - \beta_1) \left( \sum_{k=1}^{\lfloor T/r \rfloor} \sum_{t=(k-1)r}^{kr-1} \mathbb{E} \|\theta_t - \theta^\star\|^2 + \sum_{t=\lfloor T/r \rfloor r}^{T} \mathbb{E} \|\theta_t - \theta^\star\|^2 - \mathbb{E} \|\theta_0 - \theta^\star\|^2 \right)$$

$$\le \sum_{k=0}^{\lfloor T/r \rfloor} \left( \frac{G_\infty D_\infty^2}{\alpha} \sqrt{kr + 2} + 4c(1 - \beta_1)\mathbb{E} \|\theta_{kr} - \theta^\star\|^2 \right) + \sum_{k=1}^{\lfloor T/r \rfloor} \frac{dG_\infty D_\infty^2}{\alpha} \sqrt{kr - 1}$$

$$+ \frac{dG_\infty D_\infty^2 \sqrt{T}}{\alpha} + \frac{\alpha(1 + \beta_1)}{(1 - \beta_1)(1 - \delta)\sqrt{1 - \beta_2}} \sum_{k=1}^{\lfloor T/r \rfloor} \sum_{i=1}^{d} \mathbb{E} \|g_{(k-1)r+1:kr-1,i}\| \sqrt{\sum_{t=(k-1)r+1}^{kr-1} \frac{1}{t}}$$

$$+ \frac{\alpha(1 + \beta_1)}{(1 - \beta_1)(1 - \delta)\sqrt{1 - \beta_2}} \sum_{i=1}^{d} \mathbb{E} \|g_{\lfloor T/r \rfloor r+1:T,i}\| \sqrt{\sum_{t=\lfloor T/r \rfloor r+1}^{T} \frac{1}{t}}$$

$$+ G_\infty D_\infty^2 \sum_{k=1}^{\lfloor T/r \rfloor} \sum_{t=(k-1)r+2}^{kr-1} \frac{\beta_{1t}}{\alpha_t} + G_\infty D_\infty^2 \sum_{t=\lfloor T/r \rfloor r+2}^{T} \frac{\beta_{1t}}{\alpha_t}$$

$$\le \sum_{k=0}^{\lfloor T/r \rfloor} \left( \frac{G_\infty D_\infty^2}{\alpha} \sqrt{kr + 2} + 4c(1 - \beta_1)\mathbb{E} \|\theta_{kr} - \theta^\star\|^2 \right) + \sum_{k=1}^{\lfloor T/r \rfloor} \frac{dG_\infty D_\infty^2}{\alpha} \sqrt{kr - 1}$$

$$+ \frac{dG_\infty D_\infty^2 \sqrt{T}}{\alpha} + \frac{\alpha(1 + \beta_1)}{(1 - \beta_1)(1 - \delta)\sqrt{1 - \beta_2}} \sum_{k=1}^{\lfloor T/r \rfloor} \sum_{i=1}^{d} \mathbb{E} \|g_{(k-1)r+1:kr-1,i}\| \sqrt{\sum_{t=(k-1)r+1}^{kr-1} \frac{1}{t}}$$

$$+ \frac{\alpha(1 + \beta_1)}{(1 - \beta_1)(1 - \delta)\sqrt{1 - \beta_2}} \sum_{i=1}^{d} \mathbb{E} \|g_{\lfloor T/r \rfloor r+1:T,i}\| \sqrt{\sum_{t=\lfloor T/r \rfloor r+1}^{T} \frac{1}{t}} + G_\infty D_\infty^2 \sum_{t=1}^{T} \frac{\beta_{1t}}{\alpha_t}.$$

We can bound the term $G_\infty D_\infty^2 \sum_{t=1}^T \frac{\beta_{1t}}{\alpha_t}$ by Lemma 3. Next, we bound another key term in the above inequality. We first observe that $\forall k \geq 2, \forall i \in [d]$,

$$
\left\|g_{(k-1)r+1:kr-1,i}\right\| \sqrt{\sum_{t=(k-1)r+1}^{kr-1} \frac{1}{t}} \overset{(i)}{\leq} \left\|g_{(k-1)r+1:kr-1,i}\right\| \sqrt{\sum_{t=(k-1)r+1}^{kr-1} \frac{1}{t}} + |g_{kr,i}| \sqrt{\frac{1}{kr}}
$$
$$
\overset{(ii)}{\leq} \left\|g_{(k-1)r+1:kr,i}\right\| \sqrt{\sum_{t=(k-1)r+1}^{kr} \frac{1}{t}}, \tag{21}
$$

where (i) holds due to $|g_{t,i}| \sqrt{\frac{1}{t}} > 0$ and (ii) follows from the Cauchy-Schwarz inequality. Then we have

$$
\sum_{k=1}^{\lfloor T/r \rfloor} \sum_{i=1}^{d} \left\|g_{(k-1)r+1:kr-1,i}\right\| \sqrt{\sum_{t=(k-1)r+1}^{kr-1} \frac{1}{t}} + \sum_{i=1}^{d} \left\|g_{\lfloor T/r \rfloor r+1:T,i}\right\| \sqrt{\sum_{t=\lfloor T/r \rfloor r+1}^{T} \frac{1}{t}}
$$
$$
\overset{(i)}{\leq} \sum_{k=1}^{\lfloor T/r \rfloor} \sum_{i=1}^{d} |g_{kr,i}| \sqrt{\frac{1}{kr}} + \sum_{k=1}^{\lfloor T/r \rfloor} \sum_{i=1}^{d} \left\|g_{(k-1)r+1:kr-1,i}\right\| \sqrt{\sum_{t=(k-1)r+1}^{kr-1} \frac{1}{t}}
$$
$$
+ \sum_{i=1}^{d} \left\|g_{\lfloor T/r \rfloor r+1:T,i}\right\| \sqrt{\sum_{t=\lfloor T/r \rfloor r+1}^{T} \frac{1}{t}}
$$
$$
= \sum_{k=1}^{\lfloor T/r \rfloor} \sum_{i=1}^{d} \left( \left\|g_{(k-1)r+1:kr-1,i}\right\| \sqrt{\sum_{t=(k-1)r+1}^{kr-1} \frac{1}{t}} + |g_{kr,i}| \sqrt{\frac{1}{kr}} \right)
$$
$$
+ \sum_{i=1}^{d} \left\|g_{\lfloor T/r \rfloor r+1:T,i}\right\| \sqrt{\sum_{t=\lfloor T/r \rfloor r+1}^{T} \frac{1}{t}}
$$
$$
\overset{(ii)}{\leq} \sum_{k=1}^{\lfloor T/r \rfloor} \sum_{i=1}^{d} \left\|g_{(k-1)r+1:kr,i}\right\| \sqrt{\sum_{t=(k-1)r+1}^{kr} \frac{1}{t}} + \sum_{i=1}^{d} \left\|g_{\lfloor T/r \rfloor r+1:T,i}\right\| \sqrt{\sum_{t=\lfloor T/r \rfloor r+1}^{T} \frac{1}{t}}
$$
$$
= \sum_{i=1}^{d} \left( \sum_{k=1}^{\lfloor T/r \rfloor} \left\|g_{(k-1)r+1:kr,i}\right\| \sqrt{\sum_{t=(k-1)r+1}^{kr} \frac{1}{t}} + \left\|g_{\lfloor T/r \rfloor r+1:T,i}\right\| \sqrt{\sum_{t=\lfloor T/r \rfloor r+1}^{T} \frac{1}{t}} \right)
$$
$$
\overset{(iii)}{\leq} \sum_{i=1}^{d} \left\|g_{1:T,i}\right\| \sqrt{\sum_{t=1}^{T} \frac{1}{t}},
$$

where (i) follows from $|g_{kr,i}| \sqrt{\frac{1}{kr}}, \forall k \geq 1, \forall i \in [d]$, (ii) follows from Equation (21) and (iii) holds due to the Cauchy-Schwarz inequality. Then we have

$$2c(1-\beta_1)\sum_{t=1}^{T}\mathbb{E}\left\|\theta_t-\theta^\star\right\|^2$$

$$\leq \sum_{k=0}^{\lfloor T/r\rfloor}\left(\frac{G_\infty D_\infty^2}{\alpha}\sqrt{kr+2}+4c(1-\beta_1)\mathbb{E}\left\|\theta_{kr}-\theta^\star\right\|^2\right)+\sum_{k=1}^{\lfloor T/r\rfloor}\frac{dG_\infty D_\infty^2}{\alpha}\sqrt{kr-1}$$

$$+\frac{dG_\infty D_\infty^2\sqrt{T}}{\alpha}+\frac{\alpha(1+\beta_1)}{(1-\beta_1)(1-\delta)\sqrt{1-\beta_2}}\sum_{k=1}^{\lfloor T/r\rfloor}\sum_{i=1}^{d}\mathbb{E}\left\|g_{(k-1)r:kr-1,i}\right\|\sqrt{\sum_{t=(k-1)r}^{kr-1}\frac{1}{t}}$$

$$+\frac{\alpha(1+\beta_1)}{(1-\beta_1)(1-\delta)\sqrt{1-\beta_2}}\sum_{i=1}^{d}\mathbb{E}\left\|g_{\lfloor T/r\rfloor r:T,i}\right\|\sqrt{\sum_{t=\lfloor T/r\rfloor r}^{T}\frac{1}{t}}+G_\infty D_\infty^2\sum_{t=1}^{T}\frac{\beta_{1t}}{\alpha_t}$$

$$\leq \sum_{k=0}^{\lfloor T/r\rfloor}\left(\frac{G_\infty D_\infty^2}{\alpha}\sqrt{kr+2}+4c(1-\beta_1)\mathbb{E}\left\|\theta_{kr}-\theta^\star\right\|^2\right)+\sum_{k=1}^{\lfloor T/r\rfloor}\frac{dG_\infty D_\infty^2}{\alpha}\sqrt{kr-1}$$

$$+\frac{dG_\infty D_\infty^2\sqrt{T}}{\alpha}+\frac{\alpha(1+\beta_1)}{(1-\beta_1)(1-\delta)\sqrt{1-\beta_2}}\sum_{i=1}^{d}\mathbb{E}\left\|g_{1:T,i}\right\|\sqrt{\sum_{t=1}^{T}\frac{1}{t}}+G_\infty D_\infty^2\sum_{t=1}^{T}\frac{\beta_{1t}}{\alpha_t}$$

$$\overset{(i)}{\leq} \sum_{k=0}^{\lfloor T/r\rfloor}\left(\frac{G_\infty D_\infty^2}{\alpha}\sqrt{kr+2}+4c(1-\beta_1)\mathbb{E}\left\|\theta_{kr}-\theta^\star\right\|^2\right)+\sum_{k=1}^{\lfloor T/r\rfloor}\frac{dG_\infty D_\infty^2}{\alpha}\sqrt{kr-1}$$

$$+\frac{dG_\infty D_\infty^2\sqrt{T}}{\alpha}+\frac{\alpha(1+\beta_1)\sqrt{d(1+\log T)}}{(1-\beta_1)(1-\delta)\sqrt{1-\beta_2}}\sum_{i=1}^{d}\mathbb{E}\left\|g_{1:T,i}\right\|+\frac{\beta_1 G_\infty D_\infty^2}{\alpha(1-\lambda)^2},$$

where (i) follows from Lemma 2 and Lemma 3.

Finally, applying the Jensen's inequality and the above bound, we obtain

$$\mathbb{E}\left\|\theta_{out}-\theta^\star\right\|^2$$

$$\leq \frac{1}{T}\sum_{t=1}^{T}\mathbb{E}\left\|\theta_t-\theta^\star\right\|^2$$

$$\leq \frac{1}{T}\sum_{k=0}^{\lfloor T/r\rfloor}\left(\frac{G_\infty D_\infty^2}{2c\alpha(1-\beta_1)}\sqrt{kr+2}+2\mathbb{E}\left\|\theta_{kr}-\theta^\star\right\|^2\right)+\frac{1}{T}\sum_{k=1}^{\lfloor T/r\rfloor}\frac{dG_\infty D_\infty^2}{2c\alpha(1-\beta_1)}\sqrt{kr-1}$$

$$+\frac{dG_\infty D_\infty^2\sqrt{T}}{2c\alpha(1-\beta_1)}+\frac{\alpha(1+\beta_1)\sqrt{d(1+\log T)}}{2c(1-\beta_1)^2(1-\delta)\sqrt{1-\beta_2}}\sum_{i=1}^{d}\mathbb{E}\left\|g_{1:T,i}\right\|+\frac{\beta_1 G_\infty D_\infty^2}{2c\alpha(1-\beta_1)(1-\lambda)^2},$$

which concludes the proof.

