# OpenReview forum: "CAN ALTQ LEARN FASTER: EXPERIMENTS AND THEORY"
_ICLR.cc/2020/Conference — Reject_

### Official Review · AnonReviewer3 · 2019-10-16
**Official Blind Review #3**

**Rating:** 3

**Review:**

This paper is well-written and it provides a convergence result for traditional Q-learning, with linear function approximation, when using an Adam-like update (AMSGrad). It does the same for a variation of this algorithm where the momentum-like term is reset every now and then. This second result is not that exciting as it ends up concluding that the “best way” to converge with such an approach is by resetting the momentum-like term rarely. That being said, it is still interesting to have such theoretical result. On the empirical side, this paper evaluates the traditional Q-learning algorithm with non-linear function approximation (through a neural network), using Adam (and AdamR) while not using a target network, in both an LQR problem and a subset of the Atari games. The empirical results are not necessarily that convincing and there are important details missing. I’m willing to increase my score if my concerns w.r.t. the empirical validation are addressed since this paper presents a potentially interesting theoretical result with Adam, which is so prominent in the literature nowadays.

With respect to the empirical analysis, it is not clear to me why only 23 Atari games were used. The traditional answer often revolves around limited computational resources, but for this set of experiments it is particularly concerning. *How were these 23 games chosen?* The reason I ask is that DQN, without a target network, does work in approximately half of the Atari games. In the other games DQN presents instability and it doesn’t succeed. I wonder if the proposed approach would actually be stable throughout the whole set of Atari games. Moreover, it is not clear to me *why such a small buffer size was used*. As acknowledged in the paper: “Considering we use a smaller buffer size than common practice, DQN is not consistently showing improved return over all tested games”. It seems to me that DQN was evaluated in a suboptimal parameter setting, so it is not clear to me how relevant the claims that the evaluated algorithm outperforms DQN are.

The results itself, which are summarized in Figure 2 and in the Appendix, in Figure 3, are not that convincing (I would also like to see a table with the raw numbers at the end, as it is often done). In Figure 2, as acknowledged in the paper, “AltQ-Adam” has a huge variance at the end (raising concerns about its stability, as I previously mentioned). “AltQ-AdamR” does seem to be more stable but the confidence intervals (or standard deviation?) overlap (red and blue). Is there a significant difference? Looking at Figure 3 in the Appendix, the individual learning curves are not that convincing as well. In 12 out of 23 games DQN outperforms the other methods or there is pretty much no difference in performance between them (i.e., Alien, Assault, Asteroids, Bowling, CrazyClimber, FishingDerby, JamesBond, Pitfall, Pong, QBert, Robotank, Tutankham). In 11 out of 23 games there’s a clear advantage of AltQ-Adam or AltQ-AdamR (i.e., Amidar, Asterix, BeamRider, Enduro, DemonAttack(?), DoubleDunk, Gopher, Gravitar, Seaquest, SpaceInvaders, Tennis). Consequently, although AltQ seems better than DQN, the difference is not that big. Importantly, some crucial details are missing (at least I couldn’t find them). *How many seeds were used in each game?* One seed is way too little since the performance difference in Tennis could be explained by “luck” in the random action selection process. Which Atari version was used? The deterministic one or the stochastic one (Machado et al., 2018)?

With respect to presentation, I don’t understand why Q-learning is being called AltQ. AltQ, with linear function approximation, is exactly what Q-learning does. *Introducing a new name for an old algorithm is very distracting* and it actually might reduce the impact of the paper, since more people can relate to Q-learning than to AltQ, which sounds as a newly introduced algorithm. Some details are not clear in Algorithms 1 and 2. *Is there a minibatch? That’s not represented in these algorithms, it it? Would it be fair to say that AltQ is _online_ Q-learning? I don’t think so since it uses a minibatch uniformly sampled from the experience replay buffer. It would be beneficial to have a paragraph explicitly discussing what are the differences between AltQ and DQN. Is it only dropping the target network?* Finally, there are some typos (e.g., “Nestrov”) and the references could be improved. Specifically, I’d recommend the authors to not use citations as nouns (e.g., “which is justified in (Duchi et al., 2011).”) and to cite Bellemare et al. (2013) instead of (or with) Brockman et al. (2016) when referring to the Atari games.


References:

Marc G. Bellemare, Yavar Naddaf, Joel Veness, Michael Bowling: The Arcade Learning Environment: An Evaluation Platform for General Agents. J. Artif. Intell. Res. 47: 253-279 (2013)

Marlos C. Machado, Marc G. Bellemare, Erik Talvitie, Joel Veness, Matthew J. Hausknecht, Michael Bowling: Revisiting the Arcade Learning Environment: Evaluation Protocols and Open Problems for General Agents. J. Artif. Intell. Res. 61: 523-562 (2018)


---

>>> Update after rebuttal: I stand by my score after the rebuttal.

This paper has several presentation issues that need to be addressed before publication. The theoretical result does make it an interesting contribution, but the empirical section weakens the paper quite a lot. The paper is trying to show that the proposed approach is more stable but the smaller replay buffer size is not satisfying, since it is a setting which no one uses. Moreover, it is not possible to claim that a method presents lower variance when all one has are to 2 seeds to back up those claims. I don't think "limited computational capability and cost" is a good justification in that case.

Finally, the term AltQ is distracting and misleading. The authors didn't demonstrate any intention of changing this name, but if they decide to submit this paper to a next conference, with stronger empirical evidence to back up the paper claims, I recommend them to drop the AltQ learning name as well.


**Experience Assessment:**

I have published in this field for several years.

**Review Assessment: Checking Correctness Of Derivations And Theory:**

I did not assess the derivations or theory.

**Review Assessment: Checking Correctness Of Experiments:**

I carefully checked the experiments.

**Review Assessment: Thoroughness In Paper Reading:**

I read the paper thoroughly.

---

> ### Author Response · Authors · 2019-11-15
> **Response to Review #3**
>
> Thanks for your time and the valuable comments! We address the raised concerns as follows.
>
> 1 - Q：How were these 23 games chosen?
>
> A：Limited by the computational capability and cost, we did not evaluate the complete set of Atari games. The selection of the 23 games is completely random.
>
> 2 - Q：why such a small buffer size was used
>
> A：Limited by the computational capability and cost, we use a smaller buffer size than normal practice. However, this buffer size remains the same for all algorithms being evaluated.
>
> 3 - Q：I would also like to see a table with the raw numbers at the end, as it is often done. In Figure 2, as acknowledged in the paper, “AltQ-Adam” has a huge variance at the end (raising concerns about its stability, as I previously mentioned). “AltQ-AdamR” does seem to be more stable but the confidence intervals (or standard deviation?) overlap (red and blue). Is there a significant difference?
>
> A：We have added the table of raw numbers in the revision (Appendix, Table 3). Consider the best empirical return throughout the training process, DQN outperforms the two AltQ algorithms on 3 out of 23 games, AltQ-Adam achieves the best score on 13 out of 23 games and AltQ-AdamR performs the best on 7 out of 23 games.
>
> The standard deviation of the normalized score by the end of the training is 44.51 for AltQ-AdamR and 114.18 for AltQ-Adam. It is actually a significant difference. Furthermore, if we consider having a final score that is smaller or equal to the start score as a learning failure, DQN fails the learning in 5 out of 23 games (Asteroids, DoubleDunk, Gravitar, Pitfall, Tennis), AltQ-Adam fails in 3 out of 23 games (Amidar, Assault, Asteroids) and AltQ-AdamR does not fail in any of the tasks. This also indicates the stability advantage of the AltQ-AdamR algorithm.
>
> 4 - Q：Looking at Figure 3 in the Appendix, the individual learning curves are not that convincing as well. Consequently, although AltQ seems better than DQN, the difference is not that big.
>
> A：The vanilla AltQ algorithm (or Q-learning) has been considered being slow and unstable in the previous research. The improvement from AltQ to AltQ-Adam, while seemingly marginal, is still of interest to rectify the formal misunderstanding of this method in practice.
> Also, as we have mentioned in the last response, AltQ-AdamR archives the most stable training among all three methods. Furthermore, if compared with DQN, AltQ-AdamR achieves an at least on-par performance with DQN on 22 out of 23 games (only loses to DQN on Robotank).
>
> 5 - Q: How many seeds were used in each game?
>
> A: We had 10 random seeds for the LQR experiment and 2 random seeds for each Atari Game. It is mentioned in the appendix (A.2).
>
> 6 - Q: Which Atari version was used? The deterministic one or the stochastic one (Machado et al., 2018)?
>
> A: The stochastic one.
>
> 7 - Q: With respect to presentation, I don’t understand why Q-learning is being called AltQ. AltQ, with linear function approximation, is exactly what Q-learning does. *Introducing a new name for an old algorithm is very distracting* and it actually might reduce the impact of the paper, since more people can relate to Q-learning than to AltQ, which sounds as a newly introduced algorithm. “
>
> A: We agree with the reviewer that the AltQ is indeed the vanilla Q-learning. In our study, we use this term to clarify the difference from the algorithm with target network like DQN we need to compare with. The name “alternating” Q-learning emphasizes the update manner of the algorithm in practice where one “performs the update by taking one step temporal target update and one step parameter learning in an alternating fashion”.
>
> 8 - Q: Some details are not clear in Algorithms 1 and 2. Is there a minibatch? That’s not represented in these algorithms, it it? Would it be fair to say that AltQ is _online_ Q-learning? I don’t think so since it uses a minibatch uniformly sampled from the experience replay buffer. It would be beneficial to have a paragraph explicitly discussing what are the differences between AltQ and DQN. Is it only dropping the target network?
>
> A: The LQR experiment uses the single sample for update and the Atari game does use a minibatch with experience replay. We agree that we should state the algorithm in its specific form and will make the change in the updated revision. As for the difference between AltQ and DQN, it is only the absence of the target network.
>
> 9 - “Finally, there are some typos (e.g., “Nestrov”) and the references could be improved. Specifically, I’d recommend the authors to not use citations as nouns (e.g., “which is justified in (Duchi et al., 2011).”) and to cite Bellemare et al. (2013) instead of (or with) Brockman et al. (2016) when referring to the Atari games.”
>
> A: We have addressed the comments in the revision.

---

### Official Review · AnonReviewer2 · 2019-10-24
**Official Blind Review #2**

**Rating:** 3

**Review:**

This paper describes a method to improve the AltQ algorithm (which is typically unstable and inefficient) by using a combination of an Adam optimizer and regularly restarting the internal parameters of the Adam optimizer. The approach is evaluated on both a synthetic problem and on Atari games.

The core of the approach (simply replacing the optimizer with Adam) is relatively simple and the restarts seem mostly to improve variance rather than return over the vanilla Adam approach. It's hard to see what additional value the convergence analysis provides over the AMSGrad convergence analysis. Especially when the convergence rate appears to be the same for AltQ-AMSGrad and AltQ-AMSGradR for large r. Overall, it seems the approach of using Adam for the optimizer in AltQ seems to be too trivial an improvement and the difference between the Adam with restarts and Adam without restarts also seems to be relatively insignificant when looking at normalized return on Atari. They also both have increasing variance near the end of training in figure 2, much larger than that of DQN. Another downside of this work is that the convergence analysis is done on AMSGrad instead of Adam which the experimental results are based on, why were experiments not done with AltQ-AMSGrad?

Other comments:
In section 4.1: "to help preventing" -> "to help prevent"
Bottom of page 5: "most well-performed" -> "most well-performing"
Above eq 8: "step stone" -> "stepping stone"

================================================================================================
Update after rebuttal:

Thanks for the response. I will stand by my score. I still find it odd that no experiments are done with AltQ-AMSGrad since the convergence analysis was done for this algorithm. After all, the results of AltQ-Adam might be very different to AltQ-AMSGrad, which puts into question how relevant that convergence analysis is to the experimental results.

Finally, I agree with the other reviewers that AltQ is a misleading name for this algorithm and recommend the paper use a standard name for the algorithm.

**Experience Assessment:**

I do not know much about this area.

**Review Assessment: Checking Correctness Of Derivations And Theory:**

I did not assess the derivations or theory.

**Review Assessment: Checking Correctness Of Experiments:**

I assessed the sensibility of the experiments.

**Review Assessment: Thoroughness In Paper Reading:**

I read the paper at least twice and used my best judgement in assessing the paper.

---

> ### Author Response · Authors · 2019-11-15
> **Response to Review #2**
>
> Thanks for your time and the valuable comments! We address the raised concerns as follows.
>
> 1- Q: The core of the approach (simply replacing the optimizer with Adam) is relatively simple and the restarts seem mostly to improve variance rather than return over the vanilla Adam approach. It's hard to see what additional value the convergence analysis provides over the AMSGrad convergence analysis. Especially when the convergence rate appears to be the same for AltQ-AMSGrad and AltQ-AMSGradR for large r. “
>
> A: Although it seems to be a natural idea to incorporate Adam to Q-learning, we have not seen any previous work that uses Adam to AltQ type of algorithms and formally compare their performance to DQN. As a matter of fact, the AltQ algorithm has been considered slow and unstable in previous research. Our study first rectifies such a misunderstanding of the AltQ algorithm, and empirically demonstrates that by applying as simple schemes as Adam update and Restart, the performance can be significantly improved and even better than the relatively complicated DQN.
>
> On the theoretical side, our contribution mainly lies in providing the first convergence analysis for Q-learning with Adam type of schemes. The technical proofs are nontrivial, and we develop advanced techniques to deal with the challenge due to the momentum terms, which is very different from the existing proofs. The matching convergence rates for the two algorithms only imply that they have the same nature of the convergence rate, which is quite common in optimization. This fact should not diminish the significance of the theoretical guarantee.
>
> 2 - Q: Overall, it seems the approach of using Adam for the optimizer in AltQ seems to be too trivial an improvement and the difference between Adam with restarts and Adam without restarts also seems to be relatively insignificant when looking at normalized return on Atari. They also both have increasing variance near the end of training in figure 2, much larger than that of DQN.
>
> A: The standard deviation by the end of the training is 28.39 for DQN, 44.51 for AltQ-AdamR and 114.18 for AltQ-Adam. It is true that the DQN’s overall variance is still the smallest, but the higher variance of AltQ-AdamR in comparison with DQN is mainly caused by the better performance achieved on some of the games. In particular, AltQ-AdamR achieves a final score that is at least on par with DQN in 22 out of 23 games (other than Robotank).
>
> Furthermore, if we consider having a final score that is smaller or equal to the start score as a learning failure, DQN fails the learning in 5 out of 23 games (Asteroids, DoubleDunk, Gravitar, Pitfall, Tennis), AltQ-Adam fails in 3 out of 23 games (Amidar, Assault, Asteroids) and AltQ-AdamR does not fail in any of the tasks. More importantly, our algorithms can be implemented under limited resources and achieve good performance, which we believe is inspiring in practice.
>
> 3 - Q: Another downside of this work is that the convergence analysis is done on AMSGrad instead of Adam which the experimental results are based on, why were experiments not done with AltQ-AMSGrad?
>
> A: Our study follows the following logic. We are interested in the performance of Adam, since Adam is a prevailing optimizer and it is important to see how such a scheme can improve the performance of the AltQ algorithm. On the theoretical side, as we stated in the paper, Adam has been shown to not converge in the literature, and thus AMSGrad is a widely used variant to theoretically understand the convergence behavior of this class of algorithms.
>
> 4 - Other comments:
> In section 4.1: "to help preventing" -> "to help prevent"
> Bottom of page 5: "most well-performed" -> "most well-performing"
> Above eq 8: "step stone" -> "stepping stone"
>
> A: We have addressed the comments in the revision.

---

### Official Review · AnonReviewer1 · 2019-10-25
**Official Blind Review #1**

**Rating:** 1

**Review:**

This paper claims to propose a method to train q-based agents that use “alternating” Q-learning. However, the alternating approach given in the paper appears to be the normal Bellman update implemented in most versions of DQN. Furthermore, the citation given for AltQ (Mnih et al. 2016) makes no mention of the term “Alternating Q learning”.

The novelty here would be that the authors propose incorporating an Adam-like optimizer and periodically resetting the ADAM parameters. I would not consider using Adam to be sufficiently novel for publication in this venue, and the results from using parameter resetting are not so spectacular or convincing that they qualify, either. Since no ablations are given, I suspect some of the improvement could have come from just using Adam.

Finally, the convergence proofs given seem to hold only in the tabular case--not in the case when the Q function is an approximation. Generally, proofs only show that Q-learning converges in the tabular case. If these proofs held in the function approximation case, this would be a surprising breakthrough.

**Experience Assessment:**

I have read many papers in this area.

**Review Assessment: Checking Correctness Of Derivations And Theory:**

I assessed the sensibility of the derivations and theory.

**Review Assessment: Checking Correctness Of Experiments:**

I carefully checked the experiments.

**Review Assessment: Thoroughness In Paper Reading:**

I read the paper at least twice and used my best judgement in assessing the paper.

---

> ### Author Response · Authors · 2019-11-15
> **Response to Review #1**
>
> We thank the reviewer for the comments! Yet, we clarify some of the reviewer’s misunderstanding below.
>
> 1 - Q: This paper claims to propose a method to train q-based agents that use “alternating” Q-learning. However, the alternating approach given in the paper appears to be the normal Bellman update implemented in most versions of DQN. Furthermore, the citation given for AltQ (Mnih et al. 2016) makes no mention of the term “Alternating Q-learning”.
>
> A: We first clarify that this paper does not claim to propose a new Q-learning method. Instead, we are interested in comparing the performance of two dominating versions of Q-learning: DQN and AltQ. As we explain in our introduction, while DQN has been the dominating Q-learning algorithm used in practice, this paper pointed out that the simpler AltQ algorithm can outperform DQN by incorporating simple schemes such as Adam and parameter restart! To the best of our knowledge, the aspect and results in the paper are new and nontrivial.
>
> We adopt the name of AltQ to emphasize that the algorithm updates the target function with only one step and then evolves the parameters in an alternating manner. This is different from DQN with the target network. AltQ also runs by different names in the literature. One variant is one-step Q-learning in Mnih et al. (2016), which explains our citation.
>
> 2 - Q: The novelty here would be that the authors propose incorporating an Adam-like optimizer and periodically resetting the ADAM parameters. I would not consider using Adam to be sufficiently novel for publication in this venue, and the results from using parameter resetting are not so spectacular or convincing that they qualify, either. Since no ablations are given, I suspect some of the improvement could have come from just using Adam.
>
> A: First, although it seems to be a natural idea to incorporate Adam to Q-learning, we have not seen any previous work that uses Adam to AltQ type of algorithms and formally compare their performance to DQN.
>
> Second, to the best of our knowledge, this is the first study to use parameter restart in reinforcement learning. Here, we want to emphasize that the paper has compared between AltQ-Adam and AltQ-Adam-R through various examples (Figure 1 and 2) to demonstrate that parameter restart does bring extra benefits to the training beyond simply using Adam!
>
> In addition, the novelty of this work also lies in providing the first theoretical convergence rate of Q-learning with Adam-like update under linear function approximation.
>
> 3 - Q: Finally, the convergence proofs given seem to hold only in the tabular case--not in the case when the Q function is an approximation. Generally, proofs only show that Q-learning converges in the tabular case. If these proofs held in the function approximation case, this would be a surprising breakthrough.
>
> A: We want to clarify that our convergence proofs are NOT for the tabular cases, but indeed for function approximation. In fact, this is not something very surprising. As we mention in the related work section, “A large number of works have studied Q-learning with linear function approximation such as Bertsekas & Tsitsiklis (1996); Devraj & Meyn (2017); Zou et al. (2019); Chen et al. (2019b); Du et al. (2019), to name a few.” Our theoretical contribution mainly lies in providing the first convergence analysis for Q-learning with Adam-type schemes, where we develop advanced techniques to deal with the challenge aroused by the momentum terms, which is very different from the existing proofs.

---

### Decision · Program_Chairs · 2019-12-19

**Decision:**

Reject

**Comment:**

The reviewers attempted to provide a fair assessment of this work, albeit with varying qualifications.  Nevertheless, the depth and significance of the technical contribution was unanimously questioned, and the experimental evaluation was not considered to be convincing by any of the assessors.  The criticisms are sufficient to ask the authors to further strengthen this work before it can be considered for a top conference.